



# Estimating groundwater recharge from groundwater levels using non-linear transfer function noise models and comparison to lysimeter data

Raoul Collenteur[1], Mark Bakker[2], Gernot Klammler[3], and Steffen Birk[1]

[1]Institute of Earth Sciences, NAWI Graz Geocenter, University of Graz, Heinrichstrasse 26, 8010 Graz, Austria
[2]Water Management Department, Faculty of Civil Engineering and Geosciences, Delft University of Technology, Stevinweg 1, 2628 CN, Delft, The Netherlands
[3]JR-AquaConSol GMHB, Graz, Austria

**Correspondence:** Raoul Collenteur (raoul.collenteur@uni-graz.at)

**Abstract.** The application of non-linear transfer function noise (TFN) models using impulse response functions is explored to estimate groundwater recharge and simulate groundwater levels. A non-linear root zone model that simulates recharge is developed and implemented in a TFN model, and is compared to a more commonly used linear recharge model. An additional novel aspect of this study is the use of an autoregressive-moving average noise model so that the remaining noise fulfills

the statistical conditions to reliably estimate parameter uncertainties and compute the confidence intervals of the recharge estimates. The models are calibrated on groundwater level data observed at the Wagna hydrological research station in the southeastern part of Austria. The non-linear model improves the simulation of groundwater levels compared to the linear model. The annual recharge rates estimated with the non-linear model are comparable to the average seepage rates observed with two lysimeters. The recharges estimates from the non-linear model are also in reasonably good agreement with the

lysimeter data at the smaller time scale of recharge per 10 days. This is an improvement over the results from previous studies that used comparable methods, but only reported annual recharge rates. The presented framework requires limited input data (precipitation, potential evaporation, and groundwater levels) and can easily be extended to support applications in different hydrogeological settings than those presented here.

**1  Introduction**

Despite ongoing scientific efforts, the estimation of groundwater recharge remains a notoriously difficult task for hydrologists (e.g., Bakker et al., 2013). From the many techniques available (see, e.g., Healy and Scanlon, 2010, for an overview), methods using groundwater level observations as the primary source of information are among the most popular. This is likely due to the abundance of available groundwater level data and the simplicity of the methods (Healy and Cook, 2002). A well-known

example is the water table fluctuation (WTF) method, which only requires an estimate of the specific yield and groundwater





level data as model input. An additional advantage of the WTF method is that no assumptions are made about the actual recharge processes, for example the existence of preferential flow paths. This can also be considered a disadvantage, as no relationship between precipitation and recharge is established. This makes the method unsuitable for future projections of groundwater recharge when precipitation patterns change, for example in climate change impact studies.

In a review paper on the topic, Healy and Cook (2002) suggested that "approaches based on transfer function noise (TFN) models should be further explored" for the estimation of recharge. TFN models can be used to translate one or more input series (e.g., precipitation and potential evaporation) into an output series (e.g., groundwater levels) and have been adopted in many branches of hydrology (Hipel and McLeod, 1994). An early example of the use of these models for recharge estimation is given in Besbes and De Marsily (1984), through the deconvolution of groundwater levels with an aquifer unit hydrograph

obtained from a groundwater model. The study showed how the recharge flux can be related to rainfall by using an additional unit hydrograph for the unsaturated zone. Their proposed method required a calibrated groundwater model and a good estimate of the infiltration, making the method relatively laborious and less applicable in practice. O'Reilly (2004) developed a water-balance/transfer-function model to simulate recharge, using the WTF method to obtain recharge estimates to calibrate the model parameters.

In recent decades, the use of a specific type of TFN models using predefined response functions (von Asmuth et al., 2002) has gained popularity for the analysis of groundwater levels (Bakker and Schaars, 2019). In this method, impulse response functions are used to describe how groundwater levels react to different drivers such as precipitation, evaporation, and pumping. An important goal for these models has traditionally been to accurately describe the observed groundwater level fluctuations. For shallow water tables (up to a few meters depth) this can often be achieved using a simple linear relationship between

precipitation and evaporation (e.g., Berendrecht et al., 2003; von Asmuth et al., 2008). In a large scale case study for the Netherlands, Zaadnoordijk et al. (2019) obtained good results with this method for areas with shallow groundwater depths. For the simulation of deeper groundwater levels the linear relationship was shown to be less appropriate and non-linear models may be used to accurately simulate the groundwater levels (e.g., Berendrecht et al., 2006; Peterson and Western, 2014; Shapoori et al., 2015).

More recently, efforts have been made to explore the use of TFN models to estimate groundwater recharge, as suggested by Healy and Cook (2002). Hocking and Kelly (2016) constructed TFN models that included rainfall, evaporation, river levels, pumping, and a linear trend as explanatory variables, to isolate the contribution of rainfall to the groundwater level fluctuations. This contribution was then converted to recharge using the hydrograph fluctuation method (Viswanathan, 1984). Obergfell et al. (2019) used a linear model to estimate average diffuse recharge and obtained good annual recharge estimates when compared

to results from a chloride mass balance. Recognizing the importance of evaporation in their model setup, they constrained the parameter estimation by including the correct simulation of the seasonal behavior in the objective function. Peterson and Fulton (2019) used a non-linear TFN model that includes a soil moisture module to estimate recharge (Peterson and Western, 2014). To obtain reasonable estimates of recharge the model was constrained by comparing the modeled evaporation to the expected actual evaporation obtained using the Budyko curve. All of these studies reported annual recharge rates, but at least the latter

method could in principle also be used to obtain estimates at smaller time scales.





In this study, exploration of the use of non-linear TFN models using impulse response functions is continued to estimate groundwater recharge and improve the simulation of groundwater levels. A non-linear root zone model is developed based on a soil-water storage approach and implemented in a TFN model to simulate the (non-linear) effect of precipitation and evaporation on the groundwater levels. This study focuses on the estimation of recharge for relatively shallow groundwater

systems without capillary rise of groundwater to the unsaturated zone. The estimated recharge fluxes are compared to long-term recharge rates measured with two lysimeters located at the hydrological research site Wagna in Austria, providing a unique opportunity to evaluate the recharge estimates at smaller time scales. Additionally, this study documents the extension of the commonly used autoregressive model with a moving average part to model the residuals and obtain an approximately white noise series used for model calibration. The purpose of this study is to provide a proof-of-concept of the proposed methods

through a detailed case study for a single location. The data from the lysimeters are only used to evaluate the model results, and are not used to improve the results during model setup and calibration.

The next section provides an overview of the study site and the data used for model input and evaluation. In the third section, the methodological approach is described, starting with a brief overview of TFN modeling, followed by a description of the recharge models and ending with a description of the model calibration. The results are presented and discussed in the fourth

section, followed by a general discussion on the applicability of the methodology in the fifth section. The conclusions of this study are summarized and recommendations for future research are provided in the sixth and final section.

## 2 Study site and field data

The study site is at the hydrological research station Wagna in Styria, Austria. The site is located in an agricultural field surrounded by residential areas and all the required input time series are measured directly at the site. This includes the precip-

itation and the meteorological variables required to calculate potential evaporation. It is noted here that the term "evaporation" rather than "evapotranspiration" is used throughout this manuscript (e.g., Savenije, 2004; Miralles et al., 2020). The FAO-Penman-Monteith method is used to compute the daily grass-reference evaporation (Allen et al., 1998). Klammler and Fank (2014) and Schrader et al. (2013) showed that the estimates from this method are in good agreement with estimates obtained from a grass lysimeter that is present at the site. The average annual precipitation ($P$) and grass reference evaporation ($E_p$) in

the period 2007-2019 were 956 and 765 $\mathrm{mm\,yr^{-1}}$, respectively. The time series of both fluxes are shown in Fig. 1a and 1b.

Groundwater levels are observed with a daily time step since 1992 (see Fig. 1d, only data from 2006 onwards is shown here). The depth to water table is approximately 4 m and no capillary rise of moisture from the water table into the root zone is expected due to the existence of a coarse gravel layer at a depth of 0.50-120 $\mathrm{cm}$ (Klammler and Fank, 2014). The land surface is at 267 m above Mean Adriatic Sea Level (MASL) with little elevation differences and small hydraulic head gradients ($\pm 2.5$

m per km). The nearest drainage features are the Sulm river 1 km to the west and the larger Mur river 1.5 km to the east. Groundwater pumping for drinking water purposes occurs 500 meters north of the observation well at a rate of 240 $\mathrm{m^3 d^{-1}}$. Due to the low discharge and high conductivity of the aquifer, the effect of this pumping is assumed to be negligible at the





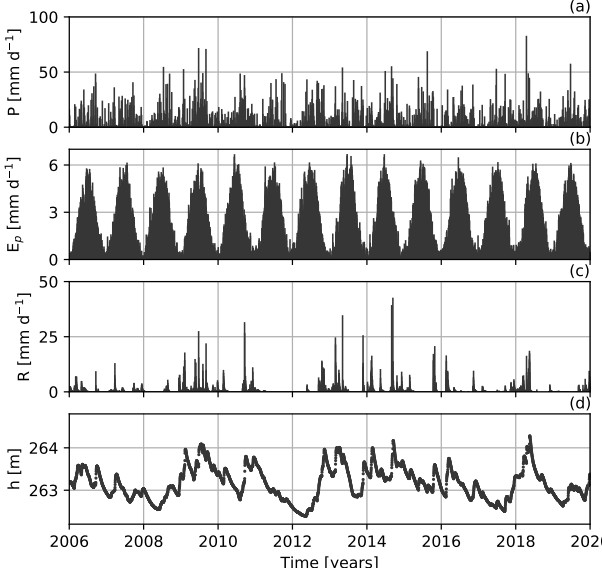

**Figure 1.** Time series of the precipitation, potential evaporation, recharge, and observed groundwater levels for the period 2006 to 2020. The recharge shown here is the average seepage measured with the two lysimeters.

study site. Given these conditions, the groundwater level fluctuations are assumed to be the exclusive result of changes in the groundwater recharge from infiltrating precipitation water.

The study site is equipped with two weighable scientific field lysimeters operated by JR-AquaConSol since 2005 (von Unold and Fank, 2008). The first lysimeter is operated under conventional farming practices (Sciencelys 1), while the second lysimeter (Sciencelys 2) was organically farmed until 2014, when it was also converted to conventional farming. A crop-rotation scheme is used for the lysimeters, with crops changing every growing season. The soils in the area are rather heterogeneous, with the thickness of the sandy loamy top layer varying greatly over short distances. The underlying sand and gravel deposits start

at a depth between 50-120 cm. Both lysimeters have an area of 1 m$^2$ and are 2 meters deep. Seepage to the groundwater is measured near the bottom of the lysimeters at 1.8 meters depth, where suction cups are installed that apply a water potential that is similar to the potential measured with tensiometers just outside the lysimeters. Both lysimeters are identical in their technical setup, and a detailed description of the lysimeters is provided in von Unold and Fank (2008) and Klammler and Fank (2014).

As the recharge is not measured at the water table itself, a certain time lag between the recharge measured with the lysimeters and the corresponding groundwater level rise exists. Only a limited time lag is expected as the ±2 m thick percolation zone consist mostly of highly conductive gravel layers. It is noted that the recharge measurements are local measurements for the area of the lysimeter, and are influenced by prevailing soil conditions, vegetation and the degree of soil sealing. The groundwater levels, measured at approximately 12 m distance from the lysimeters at Wagna test site, may also be influenced by different

recharge rates from other land-use types in the surrounding area (e.g., grassland or residential areas). As such, the measured





recharge rates from the lysimeters are – for the purpose of this paper – used as an indicative rather than an exact estimate of recharge. Considering the above, the average recharge from the two lysimeters (shown in Fig. 1c) is used in this study for the comparison with model estimates. The average recharge measured with the lysimeters is 322 mm yr$^{-1}$ over the period 2007-2019.

A number of studies have used the hydrological research site Wagna. Only the literature with a focus on recharge estimation and unsaturated flow modeling is discussed here. Fank (1999) used the water table fluctuation method to estimate groundwater recharge from observed groundwater levels and computed an average recharge of 393 mm yr$^{-1}$ over the period 1992-1996. This estimate is comparable to the 296 and 396 mm yr$^{-1}$ reported by Stumpp et al. (2009) for the two lysimeters that were operated at the site in the period 1992-2001. Stumpp et al. (2009) also applied a HYDRUS-1D model to simulate unsaturated

zone flow. Using stable isotope $\delta^{18}$O measurements it was shown that lysimeter recharge could be adequately simulated with this physically based model, although recharge peaks were generally underestimated. Groenendijk et al. (2014) documented a large comparative study of six different unsaturated zone models, where measured water content and fluxes were used to calibrate and evaluate the models. Although this study focused on nitrate leaching, the study also showed how all models had difficulties in accurately simulating the water content and fluxes observed in the current lysimeters. This was attributed to the

lack of processes such as hysteresis, preferential flow and multiple phase flow in the models. A later study using the MIKE SHE model yielded similar results (Reszler and Fank, 2016). The study concluded that the seepage and water content dynamics in the lower gravel zone inside the lysimeters could not be matched using the Richards equation and a Van Genuchten-Mualem approach, suggesting the existence of preferential flow paths below the root zone.

## 3   Methodology

### 3.1   The basic model setup

Transfer Function Noise (TFN) models are used here to translate recharge into groundwater levels. The basic model structure is:

$$h(t) = h_r(t) + d + r(t) \tag{1}$$

where $h(t)$ [L] are the observed groundwater levels, $d$ [L] is the base level of the model, $h_r(t)$ [L] is the contribution of the

recharge to groundwater level fluctuations, and $r(t)$ [L] are the model residuals. The contribution $h_r(t)$ [L] is computed by convoluting a recharge flux $R(t)$ [LT$^{-1}$] with a predefined impulse response function $\theta$ (von Asmuth et al., 2002):

$$h_r(t) = \int_{-\infty}^{t} R(\tau)\theta(t-\tau)d\tau \tag{2}$$





Following Bakker et al. (2008), a four-parameter impulse response function is used to translate the recharge flux into groundwater level fluctuations:

$$\theta_f(t) = At^{n-1}\mathrm{e}^{-t/a-ab/t} \qquad t \geq 0 \tag{3}$$

where $A$ [T$^{-n+1}$] is a scaling parameter, $a$ [T], $b$ [-], and $n$ [-] are shape parameters. For $n > 1$ the four-parameter function simulates a delayed response of the groundwater levels to recharge, while for $n \leq 1$ and $b = 0$ the groundwater levels respond instantaneously to a recharge pulse. If $n = 1$ and $b = 0$, Eq. (3) reduces to an exponential response function with only two parameters:

$$\theta_e(t) = A\mathrm{e}^{-t/a} \qquad t \geq 0 \tag{4}$$

The parameters $A$, $a$, $n$, $b$, and $d$ are estimated by fitting Eq. (1) to observed data. Depending on the hydrogeological setting and the model used to compute the recharge either a four-parameter or an exponential response function is used here to translate the recharge flux $R$ into groundwater level fluctuations. The main question that remains is how to estimate the recharge $R(t)$ from observed hydrometeorological data. The following two sections introduce the two models used in this study to compute
the recharge flux $R(t)$.

### 3.2 The Linear model

A common approach to approximate the recharge flux $R$ in Eq. (2) is a simple linear function of precipitation $P$ [LT$^{-1}$] and potential evaporation $E_p$ [LT$^{-1}$] (e.g., Berendrecht et al., 2003; von Asmuth et al., 2008):

$$R = P - fE_p \tag{5}$$

where $f$ [-] is a parameter that is calibrated. The grass-reference evaporation computed using the Penman-Monteith equation (Allen et al., 1998) is used as potential evaporation $E_p$ here. A clear interpretation of the parameter $f$ is not available. While Berendrecht et al. (2003) referred to $f$ as a crop factor, von Asmuth et al. (2008) noted that the value of $f$ "depends on the soil and land cover" instead of a single crop and also incorporates the "average reduction of the evaporation due to actual soil water shortages". Here, $f$ is referred to as the evaporation factor, following the terminology suggested by Obergfell et al. (2019).
From Eq. (5) it is clear that the flux $R$ can be negative for periods when evaporation ($fE_p$) exceeds precipitation. As Eq. (5) does not include a storage term, the temporal distribution of recharge that may result from storage in the unsaturated zone has to be captured by the impulse response function. The four-parameter response function is therefore used to translate the computed recharge into groundwater level fluctuations for the linear model. As such, the response function simulates the behavior of the





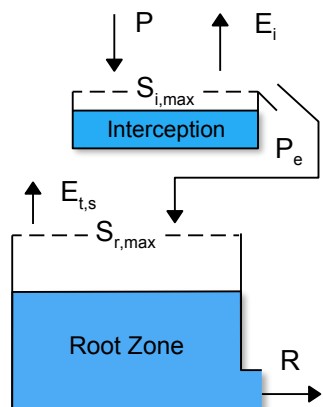

**Figure 2.** Conceptual model for the non-linear recharge model.

entire system: the root zone, unsaturated zone, and the saturated zone. In total, the linear model has six parameters to be

estimated: $A$, $n$, $a$, $b$ of the response function (Eq. (3)), the evaporation factor $f$, and the base level of the model $d$ (Eq. (1)).

### 3.3 The Non-linear model

While the linear model depends on the response function to simulate the effects of the root zone on the groundwater recharge, the non-linear model uses a soil-water storage concept to account for the temporal storage of water in the root zone. The non-linear recharge model developed here is loosely based on the FLEX conceptual modeling framework used in rainfall-runoff

modeling (Fenicia et al., 2006). The model is conceptualized as two connecting reservoirs: the first for interception and the second representing the root zone, as shown in Fig. 2. Inputs to the non-linear model are precipitation ($P$ [LT$^{-1}$]) and potential evaporation ($E_p$ [LT$^{-1}$]).

The general functioning of the model is as follows. Precipitation water is intercepted in the first reservoir until the interception capacity $S_{i,max}$ [L] is exceeded. The intercepted water can evaporate from the first reservoir as interception evaporation ($E_i$

[LT$^{-1}$]). This process forms the first barrier for precipitation to become groundwater recharge (Savenije, 2004), and creates a threshold non-linearity in the model. Precipitation exceeding the interception capacity continues as effective precipitation ($P_e$ [LT$^{-1}$]) to the root zone reservoir. From the root zone, water is evaporated through transpiration by vegetation and soil evaporation ($E_{t,s}$ [LT$^{-1}$]) or is drained to become groundwater recharge ($R$ [LT$^{-1}$]). The model is described in more detail below.

To allow the model to adjust the input potential evaporation ($E_p$) to an evaporation flux that better represents the vegetation-dependent actual evaporation, a maximum potential evaporation flux $E_{max}$ [LT$^{-1}$] is computed first:

$$E_{max} = k_v E_p \tag{6}$$



where $k_v$ [-] is a vegetation coefficient that needs to be calibrated. This approach is similar to, for example, the Ecohydrological Streamflow model developed by Viola et al. (2014). The parameter $k_v$ is interpreted as a vegetation coefficient, highlighting

the idea that the groundwater recharge may be affected by different types of vegetation instead of a single type of crop.

The water balance for the interception reservoir is:

$$\frac{\Delta S_i}{\Delta t} = P - E_i - P_e, \tag{7}$$

where

$$E_i \Delta t = \min(E_{max} \Delta t, S_i) \tag{8}$$

where $S_i$ [L] is the amount of water stored in the interception reservoir. The maximum storage capacity of the interception reservoir is determined by the parameter $S_{i,max}$ [L]. Intercepted water is evaporated from the interception reservoir, limited by the amount of maximum potential evaporation $E_{max}$ (energy-limited) or the amount of water available for evaporation $S_i$ (water-limited). Any precipitation water exceeding the interception capacity $S_{i,max}$ will continue to the root zone reservoir as effective precipitation $P_e$.

The water balance for the root zone reservoir is:

$$\frac{dS_r}{dt} = P_e - E_{t,s} - R \tag{9}$$

where $S_r$ [L] is the amount of water in the root zone reservoir, $E_{t,s}$ [LT$^{-1}$] is a combined evaporation flux constituting both soil evaporation and transpiration by vegetation, and $R$ is the recharge to the groundwater. The maximum storage capacity of the root zone reservoir is determined by the parameter $S_{r,max}$ [L]. The saturation at $t = 0$ is set to $S_r(t = 0) = 0.5 S_{r,max}$. The

evaporation flux $E_{t,s}$ is limited by the amount of water available in the root zone as follows:

$$E_{t,s} = (E_{max} - E_i) \min(1, \frac{S_r}{l_p S_{r,max}}) \tag{10}$$

where the parameter $l_p$ [-] determines at what fraction of $S_{r,max}$ the evaporation flux is limited by the availability of soil water. The relationship between the saturation of the root zone ($S_r/S_{r,max}$) and the fraction of the potential evaporation that is evaporated through the root zone ($E_{t,s}/E_{max}$) is shown in Fig. 3a. It is noted that the maximum potential evaporation is

decreased by the amount of evaporation that already took place as interception evaporation. The actual evaporation as simulated by the non-linear model is calculated as $E_a = E_{t,s} + E_i$.



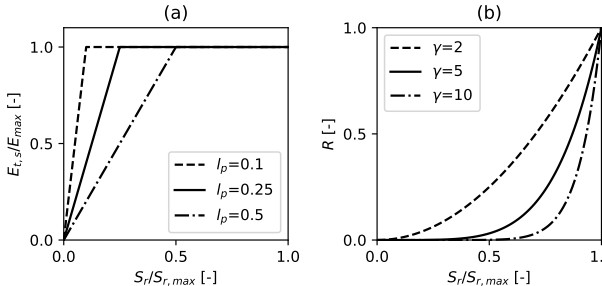

**Figure 3.** Relationships between the saturation of the root zone ($S_r/S_{r,max}$) and the evaporation (a) and the drainage from the root zone reservoir (b). The saturated hydraulic conductivity is set to $k_s = 1$.

Recharge to the groundwater $R$ is computed using Campbell's approximation for unsaturated hydraulic conductivity (Campbell, 1974).

$$R = k_s \left( \frac{S_r}{S_{r,max}} \right)^{\gamma} \tag{11}$$

where $k_s$ [LT$^{-1}$] is the saturated hydraulic conductivity and $\gamma$ [-] is a parameter that determines how non-linear this flux is with respect to the saturation of the unsaturated zone. Equation (11) reduces to the equation used in the FLEX models (Eq. (4) in Fenicia et al., 2006) when $\gamma = 1$ and is similar to that used by Peterson and Western (2014). The relationship between the saturation of the root zone and the recharge flux for different values of $\gamma$ is shown in Figure 3b.

In the preliminary phase of this study it was found that the use of an exponential response function yields similar results 210 as the four-parameter response function for the non-linear model. For reasons of model parsimony, the exponential response function (Eq. (4)) was adopted for the non-linear model to translate the recharge $R$ into groundwater levels.

In total the non-linear recharge model has 6 parameters that need to be estimated: $k_v$, $S_{i,max}$, $S_{r,max}$, $k_s$, $\gamma$, and $l_p$. Some of these parameters may be fixed to sensible values based on experience and literature values (Savenije, 2010), decreasing the number of parameters that need to be calibrated. Here, the interception capacity $S_{i,max}$ was set to 2 mm and $l_p$ was fixed to 215 0.25. The parameter $S_{r,max}$ was fixed to 250 mm (e.g., Gao et al., 2014), as it was found to have a strong correlation with $k_s$ in the preliminary phase of this study and thus hard to calibrate. This leaves a total of 6 parameters to be calibrated: $k_v$, $k_s$, and $\gamma$ of the non-linear recharge model, $A$ and $a$ of the response function (Eq. (4)), and the base level of the model $d$ (Eq. (1)).

### 3.4 The Lysimeter model

For comparison, a third model is constructed where the recharge measured with the lysimeters is used as the flux $R$ in Eq. (2). 220 Similar to the non-linear model, an exponential response function is used to translate this recharge into groundwater levels. Assuming that the recharge measured with the lysimeters is a good estimate of the (unknown) real recharge, the groundwater levels simulated with this model provide an indication of the fit that may potentially be obtained with the other models.





## 3.5 Noise modeling

The residuals $r(t)$ of TFN models applied to groundwater level data (see Eq. (1)) often show considerable autocorrelation. To
allow statistical inferences with the model (e.g., the estimation of confidence intervals of the simulated recharge) it is necessary
to transform the residuals series into a noise series that is approximately white noise. For groundwater levels time series this
generally means that the autocorrelation needs to be removed from the residuals. An autoregressive model of order one (AR(1))
is commonly used for this purpose (e.g., von Asmuth et al., 2002):

$$v(t_i) = r(t_i) - r(t_{i-1})e^{-\Delta t_i/\alpha} \qquad (12)$$

where $v$ is called the noise series here, $\Delta t_i$ is the time step between two residuals $r(t_i)$ and $r(t_{i-1})$, and $\alpha$ [T] is the AR
parameter.

In the preliminary phase of this study, the models were calibrated using daily groundwater level observations. It was found
that the noise series from these models still exhibited significant autocorrelation, despite the use of the AR(1) noise model. This
result may in fact not be that surprising, considering the slow processes governing groundwater flow systems and the model
structure used to simulate these. The former can for example be quantified by calculating the autocorrelations of the observed
groundwater levels, which in this study are higher than 0.95 for measurements up to 13 days apart and only drop below 0.5
for measurements 100 days apart. The latter is more general, where autocorrelated errors are a result from the model structure.
Errors in the input data propagate through the TFN model and are likely to result in autocorrelated errors, due to the use of a
reservoir model (Kavetski et al., 2003) and the convolution with an impulse response function.

As a practical solution, the time step between groundwater level observations was systematically increased through removal
of observations from the time series. For each increase in the interval between two measurements the models were re-calibrated
and diagnosed for autocorrelation using the Durbin-Watson (DW) test for the first time lag and the Ljung-Box test for lags up
to one year (described in the Appendix A). The results for the DW test for different time intervals are shown in Fig. 4a. A value
of DW=2 indicates that there is no autocorrelation in the noise, while DW < 2 indicates a positive autocorrelation at lag one
and DW > 2 a negative autocorrelation. While it is clearly visible in Fig. 4a that removing observations from the groundwater
level time series reduces the autocorrelation, application of the AR(1) model did not suffice for the data used in this study.

In a further attempt to remove the autocorrelation from the residuals, the AR(1) model was extended with a Moving-Average
part of order one (MA(1)) to form an ARMA(1,1) noise model as follows:

$$v(t_i) = r(t_i) - r(t_{i-1})e^{-\Delta t_i/\alpha} - v(t_{i-1})e^{-\Delta t_i/\beta} \qquad i \geq 1 \qquad (13)$$

where $\beta$ is the parameter of the moving average part of the noise model. The parameters $\alpha$ and $\beta$ are estimated during model
calibration. The first value of the noise series at $t = 0$ is set to the first value of the residuals, $v(t_0) = r(t_0)$), as it is not possible





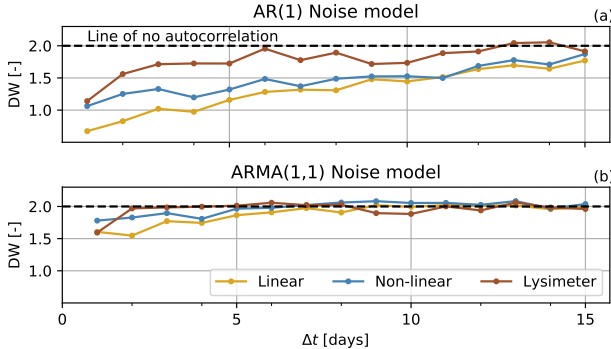

**Figure 4.** Durbin-Watson statistics for models calibrated on groundwater levels with an increasing interval ($\Delta t$) up to 15 days, using an AR(1) noise model (a) or an ARMA(1,1) noise model (b).

to compute $\upsilon(t = 0)$ from the previous residual. The MA(1) process can correct for individual shocks in the system, quickly reducing the error over one time step, whereas the AR(1) part deals with an error whose effect exponentially decreases over multiple time steps. Note that the time step $\Delta t$ in Eq. (13) may be irregular, but in this study only time series with a regular

time step are used. Additional research is necessary to make this noise model fully applicable to irregular time steps, as was done for the AR(1) model (von Asmuth and Bierkens, 2005).

Rerunning the previous analysis of the Durbin-Watson statistic for for an increasing time interval between observations using the ARMA(1,1) noise model, shows that this noise model is better capable of removing the autocorrelation at the first time lag (Fig. 4b). The autocorrelation decreases with increasing time interval and the DW value stabilizes for time intervals of around

6 days and larger. A lack of autocorrelation in the noise series at larger time lags was also confirmed using the Ljung-Box test, although the autocorrelation at lags around one year become significant for time intervals below 10 days. Based on this analysis, groundwater level time series with a 10 day time interval were used for model calibration. The final autocorrelation plots are shown in Fig. A1 in the Appendices.

### 3.6 Parameter estimation and confidence intervals

The previous three sections described the TFN models used in this study, which include a recharge model, a response function, and an ARMA(1,1) noise model. An overview of the entire TFN modeling process is shown in Fig. 5. The model parameters are estimated by fitting the simulated groundwater levels to the observed groundwater levels. The linear and non-linear models both have eight parameters that are estimated, and the lysimeter model has five parameters. A non-linear least squares approach is used here to estimate the parameters for each model simultaneously. The following objective function is used, minimizing

the sum of the squared noise:

$$F_{obj} = \sum_{i=1}^{n} \upsilon_i^2 \tag{14}$$





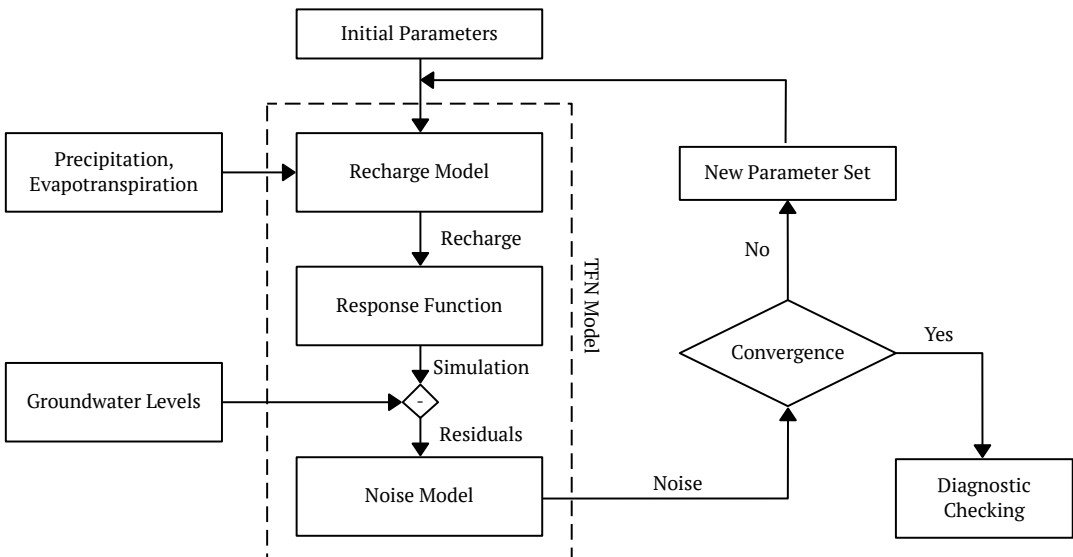

**Figure 5.** Modeling strategy as applied in this study.

Minimization of the objective function is done using the Trust Region Reflective algorithm, as implemented in Scipy's least squares method (Virtanen et al., 2020, version 1.4.0) and Lmfit (Newville et al., 2016, version 1.0). Note that this is not the default option in Lmfit. The standard errors of the parameters are computed from the covariance matrix that is estimated

during optimization. An important assumption underlying this approach is that the minimized noise series ($v$ in Eq. (13)) behaves as normally distributed white noise with no significant autocorrelation, a constant variance (homoscedastic), and a mean of zero. These assumptions were checked through visual inspection of the results and the use of various statistical tests for autocorrelation as already shown in the previous section.

The 95% confidence intervals of the simulated recharge are computed through a Monte Carlo simulation ($N$=100,000).

Parameter sets are drawn from a multivariate normal distribution computed using the estimated covariance matrix. If one of the parameters in a parameter set is outside the parameter boundaries, the set is discarded from the sample and a new parameter set is drawn. This procedure is repeated until $N$ parameter sets are available for the Monte Carlo simulation. The model is run with the $N$ different parameters sets and the 95% confidence intervals are computed from the ensemble of simulated recharge fluxes.

**3.7 Numerical and software implementation**

All models were implemented in Python code and are freely available through the open-source package Pastas (Collenteur et al., 2019, version 0.15b). The non-linear model is available under the name "Flexmodel" in the Pastas library. The non-linear recharge model is numerically solved using an explicit Euler scheme with a time step of 1 day. As TFN models are traditionally computationally inexpensive and have short computation times (in the order of seconds), special attention was paid to increase





the computation speed of the recharge model. This was achieved by using Numba, a just-in-time compiler for Python code (Lam et al., 2015, version 0.48). As a result the non-linear model has similar computation times as the linear model.

## 3.8 Goodness-of-fit metrics

Four metrics are used to evaluate the goodness-of-fits of the simulated groundwater levels and the groundwater recharge: the Mean Absolute Error (MAE), the Nash-Sutcliffe Efficiency (NSE), the coefficient of determination ($R^2$), and the Kling-Gupta

Efficiency (KGE). The MAE provides a metric for the overall model fit, while the NSE is a goodness-of-fit metric commonly used in hydrological modeling. The $R^2$ is a common error metric and is in hydrogeological TFN modeling sometimes referred to as the Explained Variance Percentage (EVP, von Asmuth et al. (2002)). The KGE is an aggregate metric and contains a correlation term, a bias term and a variability term (see Kling et al., 2012, for a more detailed discussion). The NSE, $R^2$ and KGE all have a maximum of 1, denoting a perfect fit of the model with the data. The MAE improves when moving towards

zero. All metrics are implemented in the Python package HydroStats (Roberts et al., 2018, version 0.78) that was used to compute goodness-of-fit metrics for this study.

## 4 Results & Discussion

### 4.1 Groundwater level simulations

The ten-year period 2007-2016 was used for calibration and the three year period 2017-2019 was used for model validation.

The year 2006 was used for model warm up. The use of a warm up period is especially important for the non-linear model, because the recharge flux strongly depends on the initial saturation level of the root zone. The simulated and the observed groundwater levels are shown in Fig. 6a, along with the estimated recharge fluxes (Fig. 6b and c) and the measured recharge (Fig. 6d). As the models are calibrated on groundwater level observations with a 10 day time step, only recharge rates summed over 10 day intervals are presented here. The blue shadings denote the 95% confidence intervals of the recharge estimates. The

step responses characterizing how the groundwater levels respond to a sudden unit recharge event for each of the models are shown in the inset plot at the top of Fig. 6a. The values of the calibrated parameters can be found in Table A1 in the Appendix.

All three TFN models are able to capture the major groundwater dynamics and simulate the observed groundwater levels reasonably well. For the calibration period, the non-linear model shows the best simulation of the groundwater levels as quantified by the four goodness-of-fit metrics used in this study (see Table 1). The linear model performs better than the lysimeter

model in terms of NSE and KGE metrics, but the lysimeter model shows similar or better performance according to the MAE and $R^2$, respectively. For the validation period, the differences in the metrics are not as clear, and no model outperforms the other models. For example, the lysimeter model captures the single peak in groundwater levels during the validation period better than the other models, but shows the worst simulation of the low groundwater levels that follow this peak. The linear model performs better during that period with low groundwater levels, but overestimates the low groundwater levels observed

**Figure 6.** Observed and simulated groundwater levels and the estimated and measured recharge rates. The groundwater level measurements used for calibration are shown as black dots and unused measurements are shown as gray dots. The inset plot shows the characteristic step response calibrated for each model. Note the different scale for the y-axis used for recharge computed with the linear model.





**Table 1.** Goodness-of-fit metrics for the groundwater level simulation for each model. The metrics are computed for the calibration period (2007-2016) and the validation period (2017-2019). Groundwater level measurements with a ten day time interval were to calculate these metrics, similar to the measurements used for calibration.

|  | Linear | | Non-linear | | Lysimeter | |
| --- | --- | --- | --- | --- | --- | --- |
|  | Cal. | Val. | Cal. | Val. | Cal. | Val. |
| **MAE [m]** | 0.17 | 0.14 | 0.12 | 0.13 | 0.18 | 0.18 |
| **$R^2$ [-]** | 0.76 | 0.85 | 0.86 | 0.82 | 0.78 | 0.71 |
| **NSE [-]** | 0.74 | 0.73 | 0.86 | 0.75 | 0.65 | 0.58 |
| **KGE [-]** | 0.86 | 0.75 | 0.93 | 0.76 | 0.73 | 0.82 |

at the beginning of the validation period. The non-linear model generally shows good performance, but underestimates the low groundwater levels at the end of the validation period.

While the groundwater levels simulated by the linear and non-linear models are rather similar, the groundwater recharge fluxes $R$ computed by these two models are very different (see Fig. 6). The recharge fluxes are compared to the recharge measured with the lysimeters by computing the same goodness-of-fit metrics (Table 2). The recharge flux computed by the

non-linear model shows a reasonably good fit resulting in, for example, a Kling-Gupta Efficiency of $KGE = 0.67$ for the calibration period (see also Table 2). The recharge computed by the linear model however, deviates strongly from the lysimeter recharge and often simulates negative recharge that was not measured with the lysimeters. It is concluded that the linear model should not be used to estimate groundwater recharge at this small time scale (10-day intervals), as expected. For the simulation of groundwater levels, the linear model may still be appropriate, as the difference in the recharge flux can be compensated for

by the shape of the response function. This is clearly the case here, as is visible by the differences in the step response functions shown in the inset plot in Figure 6a.

Although the linear recharge model in combination with the four-parameter response function works well to simulate most of the groundwater levels time series, the model fails under conditions where evaporation is limited by the availability of soil moisture. This occurs for example in the years 2010, 2013, and 2017, when the linear model simulates a stronger decline

in groundwater levels than was observed. These strong declines in simulated groundwater levels are caused by continued (modeled) evaporation over the summer months, resulting in negative recharge rates (as visible in Fig. 6b) and ultimately lower groundwater levels. Measurements from the lysimeters (data not shown) show that actual evaporation is only a fraction of the potential evaporation during those periods. Similar behavior for the simulation of low groundwater levels was found by Berendrecht et al. (2006), using the same linear recharge model. These results confirm that the linear recharge model should

not be used to simulate groundwater levels under such soil moisture limited conditions.

The non-linear model performs much better under such soil-moisture limited conditions and simulates almost no recharge during these periods. The non-linear model resembles the recharge behavior as measured with the lysimeters reasonably well; recharge occurring primarily as individual events, interluded by extended periods of reduced recharge. The behavior of event-based recharge was also found in other studies (Groenendijk et al., 2014; Reszler and Fank, 2016), and suggests that recharge





**Table 2.** Performance metrics for the similarity between the estimated recharge and the measured recharge, in mm per 10 days. The metrics are shown for the calibration period (2007-2016) and the validation period (2017-2019).

|  | Linear | | Non-linear | |
| --- | --- | --- | --- | --- |
|  | Cal. | Val. | Cal. | Val. |
| **MAE [mm]** | 18.71 | 14.72 | 5.80 | 4.88 |
| **$R^2$ [-]** | 0.20 | 0.17 | 0.64 | 0.48 |
| **NSE [-]** | -1.64 | -1.96 | 0.64 | 0.43 |
| **KGE [-]** | 0.27 | 0.22 | 0.67 | 0.60 |

paths are activated when a certain threshold in the soil moisture is exceeded. This non-linear response of recharge to infiltrating precipitation also becomes clear when examining the estimated values for the parameter $\gamma$, which indicates a non-linear response with a value of $\gamma = 2.91$ [-]. The results show that the use of a non-linear recharge model improves the simulation of groundwater levels at the study site, while also providing a reasonable estimate of the recharge flux $R$ at this time scale.

   It is somewhat surprising that the lysimeter model does not outperform the other two models. Three periods with deviating
groundwater levels that stand out in particular are discussed here: the low groundwater levels in 2011, the peak in 2013, and a low in 2015. As the groundwater level fluctuations are primarily the result of individual recharge events and the groundwater system has a long memory, such periods with groundwater levels deviating for a longer period of time are likely the result of errors in the quantification of individual recharge events. In 2011 almost no recharge was recorded in the lysimeters, coinciding with an underestimation of the simulated groundwater levels. From the groundwater level measurements, however, it is clear
that some recharge must have taken place, visible by temporarily stagnating and even slightly increasing groundwater levels during that period. Due to a technical issue with the lysimeters, no groundwater recharge was recorded by the lysimeters for parts of 2015, explaining the deviation in simulated groundwater levels in that year. No explanation could be found for the peak in 2013, but this may just as well be an error in the measurement of a single event, causing a long term deviation in the groundwater level simulation.

## 4.2   Annual recharge rates

   Groundwater resource managers are often interested in annual recharge rates. In this section the ability of the models to estimate recharge at this time scale is investigated. The annual recharge rates computed by the TFN models and the annual recharge measured with the lysimeters are shown in Fig. 7. The non-linear model performs better than the linear model, also shown by the descriptive statistics of the deviation [mm] between measured and estimated annual groundwater recharge rates shown
in Table 3. This is particularly true for wet years where the linear model shows large deviations (up to 239 mm yr$^{-1}$) in the annual recharge rates. The largest deviation for the non-linear model occurs during the dry year of 2011 (123 mm yr$^{-1}$). The recharge computed with the linear model has much wider confidence intervals, despite (or maybe because of) having only one calibration parameter ($f$ in Eq. (5)). This means that the practical use of the recharge estimate from the linear model may be





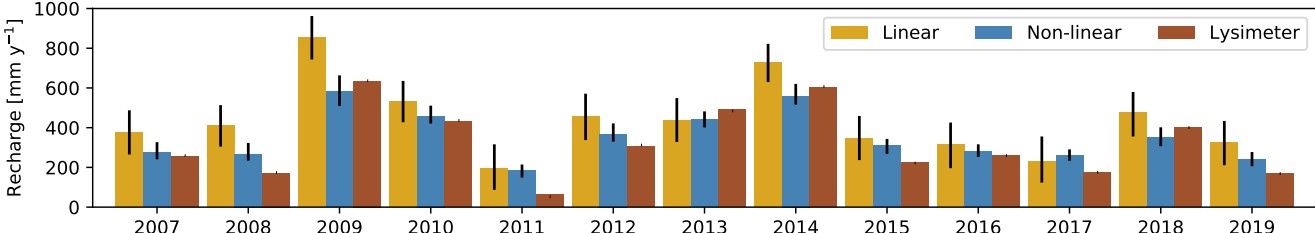

**Figure 7.** Annual recharge rates as computed by each TFN model and as measured with the lysimeters. The error bars denote the 95% confidence intervals of the recharge estimate.

**Table 3.** Descriptive statistics of the deviation (in mm) between measured and estimated annual groundwater recharge rates.

|          | Linear  | Non-linear |
|----------|---------|------------|
| **mean** | 114.96  | 29.98      |
| **min**  | -51.89  | -53.81     |
| **max**  | 238.95  | 123.42     |
| **std**  | 74.17   | 62.71      |

limited, as any analysis that uses this estimate as input data would have large uncertainties in its outcomes due to the uncertainty
in the input data. The non-linear model performs much better in this respect.

The long term average recharge (calculated for the period 2007-2019) estimated by the non-linear model ($352 \mathrm{~mm~d^{-1}}$) is much closer to the recharge measured with the lysimeters ($322 \mathrm{~mm~d^{-1}}$) than to that of the linear model ($437 \mathrm{~mm~d^{-1}}$). The overestimation of recharge by the linear model can be explained by an underestimation of evaporation that results from a low value for the evaporation factor $f$ in Eq. (5), $f = 0.69$. From the actual evaporation flux computed from the lysimeter
data (Klammler and Fank, 2014) however, it was calculated that the actual evaporation is approximately 88% of the potential evaporation (or, $f = 0.88$) for the period 2007-2019. These results confirm findings from Obergfell et al. (2019) that the factor $f$ is difficult to estimate and hampers the accurate estimation of recharge using the linear model.

An accurate estimate of evaporation is also important for recharge estimates made with the non-linear model. In Fig. 8 the annual cumulative sums of recharge and actual evaporation are shown as simulated by the non-linear model and measured
with the lysimeters (computed from 1st of Jan. to 31st of Dec.). The actual evaporation computed by the model is close to that measured with the lysimeters, averaging 81% of the potential evaporation. The vegetation coefficient $k_v$ in Eq. (6) is calibrated at $k_v = 1.48$ [-], which seems quite high at first. For most of the simulation period, however, the saturation of the root zone ($S_r / S_{r,max}$) is well below the level ($l_p = 0.25$) where evaporation from the root zone equals the potential evaporation that is left after interception evaporation, as visible in the Fig. 8c. As a result the actual evaporation simulated by the non-linear model
is still below the potential evaporation, but matches the actual evaporation measured with the lysimeters rather well.





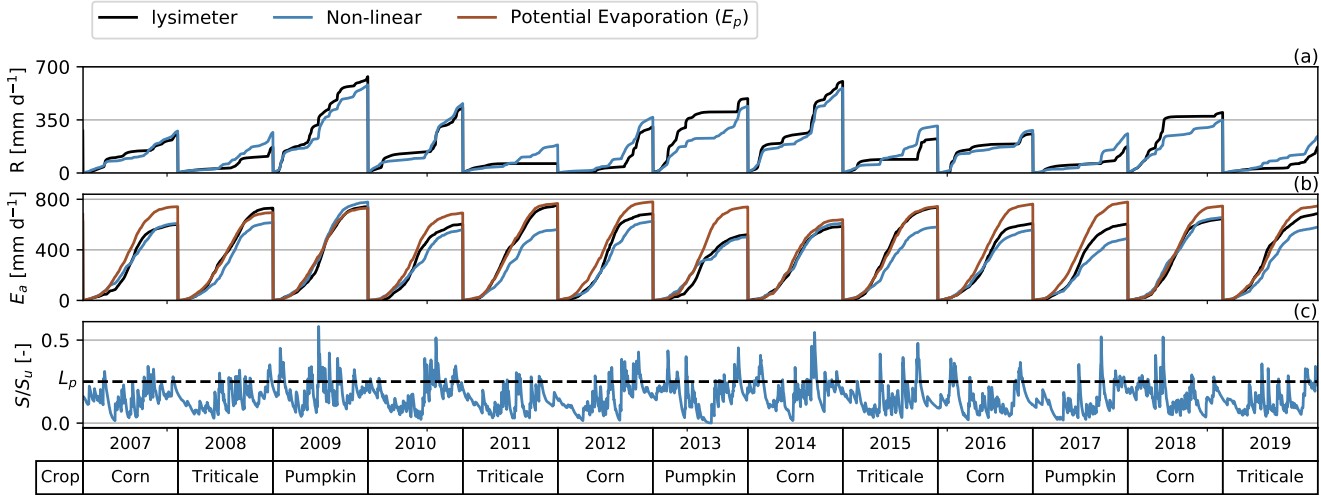

**Figure 8.** Yearly cumulative sums of recharge (a), the actual and potential evaporation (b), and the saturation of the root zone (c). The dashed line in plot (c) denotes the value of $l_p$, where the evaporation from the root zone will equal the potential evaporation.

In Fig. 8 it is visible that for years where the actual evaporation computed by the non-linear model more or less equals the actual evaporation measured with the lysimeter model, the recharge fluxes match better as well. When actual evaporation is underestimated by the model, the recharge is overestimated (see e.g., 2008, 2011, and 2015) relative to the lysimeter recharge. A probable cause for the underestimation of evaporation is the cultivation of different crops in the lysimeters during the observation period (shown in the table below the plots in Fig. 8). For example, for all years when Triticale was planted the actual evaporation was underestimated and the recharge overestimated. As grass reference evaporation was used as input data and the vegetation coefficient $k_v$ is assumed to be constant through time, the different evaporative capacities of the individual crops is not considered in the current model setup. Cultivation of different crops does not only influence the total yearly evaporation, but also the pattern in time as a result of different growing seasons and harvest times. Such effects can again be observed for Triticale, a crop that starts transpiring early in the year, visible by an earlier rise of the cumulative evaporation in years when Triticale is planted. The use of improved input data for evaporation, taking into account the impact of vegetation on this flux, may further improve recharge estimations and groundwater level simulations, particularly in agricultural areas with crop rotation schemes.

### 4.3   Parameter estimation and consistency of model output

The results presented so far are based on the calibration of the models using only 1 out of every 10 groundwater level measurements. The use of only a selection of the available groundwater level measurements during calibration allowed for a further investigation into the consistency of the modeling results, by calibrating the models to 10 different selections derived of the original time series as a type of split-sample test. This way, it is possible to assess the consistency of the estimated parameters and the impact on the simulated groundwater levels and the annual recharge estimates for this particular time series.





The resulting ensembles of groundwater level simulations and annual groundwater recharge estimates are shown in Fig. A2
       in the Appendix. The results show that both the simulated groundwater levels and the estimated recharge fluxes are consistent
       between the different calibrations for all models. This should in fact not be that surprising, considering that the time series used
       for calibration originate from the same groundwater level time series.

       What may be more surprising, however, are the differences in the estimated parameters between the 10 different calibrations
(see Fig. A3 in the Appendix). The parameter values for all models are of the same order of magnitude and model performance
       measured in $R^2$ is relatively stable, but the optimal parameter values can differ significantly from each-other between calibra-
       tions (e.g., for the non-linear model $k_s$ ranges between 100 and 250 $\mathrm{mm\ d^{-1}}$) even though the estimated confidence intervals
       overlap for the most part. The results of this split sample test raise the question of parameter identifiability. Given the similar-
       ity in simulated groundwater levels and annual recharge estimates, it is clear that different combinations of parameters yield
similar results. This split sample test shows that caution is needed when interpreting values of individual (optimal) parameters.
       Further research is necessary on the identification of parameters, for example through testing the models on large samples of
       groundwater time series (similar to, e.g., Perrin et al., 2001).

## 5    Applicability of the methodology

### 5.1    Hydrogeological setting

The presented approach was tested on relatively shallow groundwater levels (±4 m depth to the water table), for which no
       feedback between the groundwater and root zone was expected. In this setting an exponential response function was used in
       the non-linear model and the computed flux $R$ could be directly interpreted as groundwater recharge. The use of an exponential
       function may not be appropriate for deeper groundwater bodies with thicker unsaturated zones. A response function that
       accounts for this could then be used (e.g., the four-parameter response function), but the estimated flux $R$ should then be
interpreted as drainage from the root zone to the groundwater and not as recharge occurring at the water table. Peterson and
       Fulton (2019) suggested that the flux could be "averaged over a period greater than the time lag" to provide an estimate of gross
       recharge in this case. This approach was applied here for the presentation of the annual recharge rates, where also the recharge
       estimates from the linear model were considered.

       To make the methods applicable in other hydrogeological settings than those presented here, additional hydrological pro-
cesses (e.g., snow melt, surface runoff) and variables (e.g., pumping, river levels) may be included in the model. In the current
       framework, it is relatively easy to account for other variables causing groundwater level fluctuations (e.g., von Asmuth et al.,
       2008; Collenteur et al., 2019). This would allow for the estimation of recharge in hydrogeological systems where the ground-
       water level fluctuations are (possibly) not exclusively the result of recharge. Obergfell et al. (2019) already successfully tested
       this approach for groundwater levels that were also influenced by groundwater pumping. To make the recharge models appli-
cable in different settings, additional processes may be implemented in the root zone model, for example precipitation entering
       the system as snow or leaving as surface runoff before infiltrating into the soil. For ideas on how to include such processes one
       can draw from the vast number of concepts already available in conceptual rainfall-runoff modeling (e.g., Beven, 2011).





There may also be possibilities for knowledge transfer in the reverse direction, fading the boundaries between hydrologists and hydrogeologists (e.g., Staudinger et al., 2019). Groundwater levels are not often considered in conceptual rainfall-runoff

models, although it has been shown that groundwater level time series can be used to further constrain parameters in these models (e.g., Seibert, 2000). This study showed how groundwater levels may be used to calibrate the parameters of a root zone module, which is based on the conceptualization taken from a rainfall-runoff model (Fenicia et al., 2006). It would be interesting to see new attempts to constrain the parameter estimation in rainfall-runoff models using groundwater levels, in particularly using the concepts of impulse responses to improve the simulation of groundwater levels without adding many

parameters. Conversely, the results of such analyses may also help to further constrain the non-linear recharge models used in the TFN models presented here.

## 5.2 Noise modeling and quantifying uncertainties

In this study, model parameters are estimated by fitting simulated groundwater levels to observed groundwater levels. The recharge is an intermediate model result that is not calibrated for, and it is recommended to quantify the impact of parameter

uncertainties on the recharge estimates by computing their confidence intervals. To obtain reliable estimates of the parameter standard errors of in the presented framework, it is important to remove the autocorrelation from the minimized noise series by using an appropriate noise model. Here, an AR(1) did not suffice for this purpose and an ARMA(1,1) noise model was used instead. While this model was more successful in transforming the residual series into a noise time series that is approximately white noise, some autocorrelation still remained in the noise series. As a practical solution, the time interval between ground-

water level measurements was increased to 10 days by removing measurements from the time series. It should be noted here that the optimal time interval is likely to be site specific and should be investigated per individual time series. The approach shown in Section 3.5 can be helpful in determining the optimal time step size used for model calibration. The modeling of the residuals and the choice of an appropriate noise model and time interval should be considered an iterative process, as also suggested in Smith et al. (2015).

## 460 5.3 Time series requirements

The presented method requires time series of groundwater levels, precipitation, and potential evaporation as model input data. The precipitation and evaporation time series should have a regular (daily) time step to compute the recharge to the groundwater. The requirements on the groundwater level time series are less stringent, and larger or even irregular time steps between observations are allowed. To calibrate the model on irregular time series it is necessary to adapt the objective function (e.g.,

von Asmuth and Bierkens, 2005). The limited requirements on the groundwater level time series makes the method applicable to many historical time series, which often exhibit irregular time steps and data gaps. For the length of the calibration period, van der Spek and Bakker (2017) recommended a 10 to 20 year period for simulating groundwater levels with reliable credible intervals. More research is needed to determine the effect of time series length on the recharge estimation.





## 5.4 Right answers for the right reasons

The non-linear root zone model presented in this study is only one of many similar alternatives. Comparable results were obtained using a similar root zone model developed by Berendrecht et al. (2006), which is also available in the Pastas software (Collenteur et al., 2019). It is expected that other comparable non-linear model setups (e.g. the models of Peterson and Western, 2014) perform in a similar manner. Such non-linear models are better capable of simulating true system dynamics which are commonly not measured, such as groundwater recharge and actual evaporation. This suggests that the improved groundwater

level simulation is the result of a better representation of the hydrological processes, rather than merely the result of added mathematical complexity. As such, the use of non-linear root zone models in TFN models is a promising step in the effort "to get the right answers for the right reasons", as advocated by, e.g. Kirchner (2006).

## 6 Conclusions

The application of linear and non-linear transfer function noise (TFN) models using predefined impulse response functions was

explored to estimate recharge and simulate groundwater levels. The methods were tested on groundwater levels observed at the Wagna hydrological research station in the Southeastern part of Austria. A first model calculated recharge as a linear function of precipitation and evaporation, while a second model used a non-linear root zone model for this purpose. The computed recharge fluxes were compared to the average recharge flux measured with two lysimeters that are present at the research site. A third TFN model was constructed for comparison, using the lysimeter measured recharge as input data to simulate the

groundwater levels. All models were calibrated to observed groundwater levels, with the recharge as an intermediate flux that is not calibrated for in the first two models.

The results show that both the linear and non-linear TFN models are capable of simulating the groundwater levels reasonably well, with the non-linear model slightly outperforming the linear model. The use of a more complex response function was required to obtain satisfactory results with the linear model, as the response function also had to simulate the effects of the

root zone. However, this response function was not able to compensate for all hydrological conditions, and in particular during periods of low soil-moisture levels the lack of soil moisture dynamics in the linear model leads to larger errors in the simulation of the groundwater levels. These findings confirm those from other studies (Berendrecht et al., 2006; Peterson and Western, 2014) and advocate a more widespread adoption of non-linear recharge models in TFN modeling of groundwater levels. The use of such models does not necessarily imply a higher number of calibration parameters; i.e., the linear and non-linear models

used in this study have the same number of calibration parameters.

The use of a non-linear root zone model to compute the recharge in the TFN model improved the estimation of groundwater recharge significantly. For annual recharge rates it was found that the non-linear model provides good estimates with relatively small deviations from the recharge measured with the lysimeters, while the linear models shows significantly larger deviations and a structural overestimation of annual recharge rates. The non-linear model also provided reasonable estimates for recharge

summed over 10-day periods, suggesting that this model may be used to obtain recharge estimates at smaller time scales than reported so far (e.g., Obergfell et al., 2019; Peterson and Fulton, 2019). Using detailed information from the lysimeters present





at the study site, deviations in the recharge estimate could be linked to errors in the simulation of the actual evaporation, highlighting the value of field data from lysimeters and the importance of the evaporative flux in the estimation of recharge.

The methods developed in this paper were tested on a single groundwater time series and are presented as a proof-of-
concept. Additional research is needed using larger groundwater level data sets to investigate the general applicability of the method under different hydrogeological settings. The methods can also be extended to estimate recharge in settings where other hydrological stresses cause groundwater fluctuations (e.g., river levels and pumping) or when other processes (e.g., snow melt, surface runoff) influence recharge generation. To support and encourage such applications and future research, the models are included and documented in the open-source software Pastas and all scripts used for this study are made available.

*Code and data availability.*  All code used in this manuscript is available through the Python package Pastas (Collenteur et al., 2019). The data used for this manuscript is available upon request from JR-AquaConSol.

## Appendix A:  Testing for autocorrelation

Two tests are used to diagnose the minimized noise series for autocorrelation: the Durbin-Watson test and Ljung-Box test. The Durbin-Watson statistic tests the null-hypothesis that the correlation between the noise values at lag one equals zero and is
computed as follows:

$$DW = \frac{\sum_{t=2}^{n}(v_t - v_{t-1}^2)}{\sum_{t=1}^{n} v_t^2} \tag{A1}$$

where $n$ is the number of values in the noise series. The test-statistic has a range $0 \geq DW \leq 4$, where values of $DW < 2$ indicate a positive correlation and values of $DW > 2$ indicates negative autocorrelation. The Durbin-Watson test requires a constant time interval of the noise series and tests for autocorrelation at a lag of one time step.
The Ljung-Box test tests the null-hypothesis that the noise series are independently distributed for all desired time lags and is computed as follows:

$$Q = n(n+2)\sum_{k=1}^{h} \frac{\rho_k^2}{n-k} \tag{A2}$$

where $\rho_k$ is the autocorrelation at lag $k$, $h$ is the maximum lag used for calculation, and $n$ is the number of values in the noise series. A maximum time lag of $h = 36$ [-] is used here, translating to the autocorrelation for measurements that are approxi-
mately one year apart. The computed $Q$-statistic is then compared to a critical value computed from a $\chi^2_{\alpha,h-p}$ distribution with a significance level $\alpha$ and $h - p$ degrees of freedom, where $p$ is the number of parameters of the noise model. The Ljung-Box test requires a constant time step and tests for autocorrelations up to lag $h$.

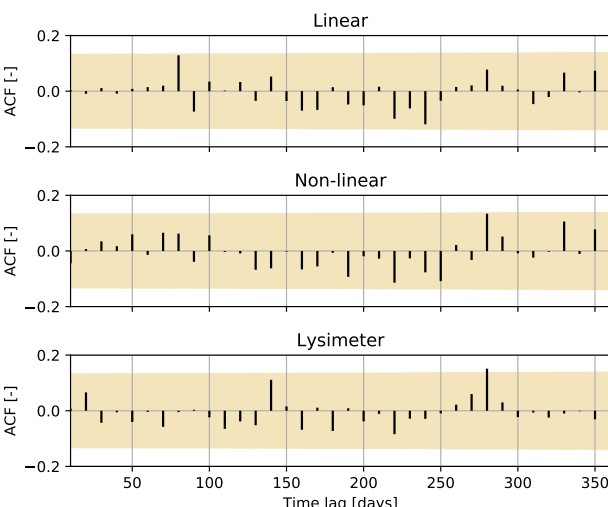

**Figure A1.** Autocorrelation graphs for all three models for lags up to one year. The shaded area shows the 99% confidence interval for the autocorrelation function.

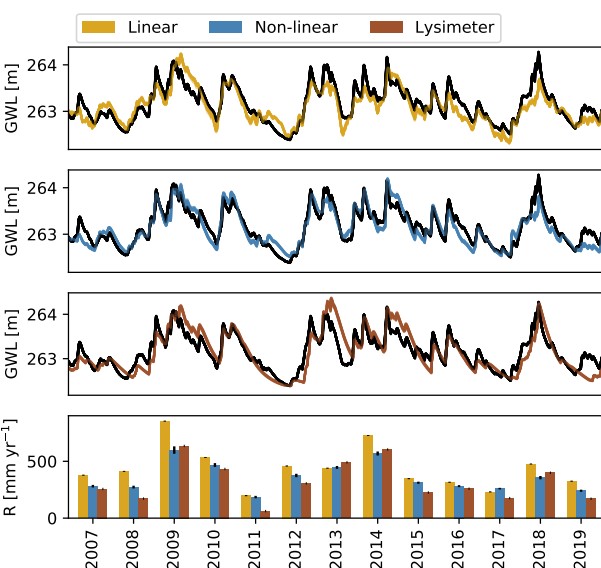

**Figure A2.** The first three plots show ensembles of 10 groundwater levels time series simulated by the three models.The bottom plot shows the mean of the estimated annual recharge rates from an ensemble of 10 models. The black whiskers show the 1.96 times the standard deviations of the ensemble of annual recharge rates.



**Figure A3.** Parameter values and their 95% confidence interval for the three models calibrated on 10 time series. Note that when possible the y-axis is shared between the different columns for comparison purposes. The bottom right subplot shows the model performance for each calibration, measured as $R^2$ between the observed and simulated groundwater levels.





**Table A1.** Calibrated parameter values for all three model configurations. The estimated standard errors of the parameters are reported between the brackets. The units from parameter $A$ depend the type of response function.

|  | Linear | Non-linear | Lysimeter |
|---|---|---|---|
| $A$ [*] | 0.58 (0.09) | 0.89 (0.09) | 1.01 (0.17) |
| $a$ [d] | 108.21 (25.49) | 116.97 (12.73) | 165.04 (30.11) |
| $b$ [d] | 0.03 (0.01) | - | - |
| $n$ [-] | 1.13 (0.10) | - | - |
| $d$ [m] | 262.41 (0.16) | 262.28 (0.09) | 262.28 (0.15) |
| $\alpha$ [d] | 93.84 (26.31) | 82.73 (20.39) | 207.23 (89.82) |
| $\beta$ [d] | 9.92 (1.42) | 10.08 (1.42) | 7.89 (1.15) |
| $f$ [-] | -0.69 (0.08) | - | - |
| $k_v$ [-] | - | 1.48 (0.17) | - |
| $\gamma$ [-] | - | 2.91 (0.30) | - |
| $k_s$ [mm d$^{-1}$] | - | 118.79 (30.95) | - |

*Author contributions.* RC and MB wrote the model code. GK provided the data and the information on the study site. RC performed the analysis and prepared the manuscript. MB and SB helped in the conceptualization of the non-linear root zone models and MB, SB and GK
reviewed different versions of the manuscript.

*Competing interests.* The authors declare that they have no conflict of interest.

*Acknowledgements.* The authors thank Willem Jan Zaadnoordijk and Markus Hrachowitz from the TU Delft for earlier work related to this manuscript. This work was funded by the Austrian Science Fund (FWF) under Research Grant W1256 (Doctoral Programme Climate Change: Uncertainties, Thresholds and Coping Strategies).



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
