# Peer review of "Estimation of groundwater recharge from groundwater levels using non-linear transfer function noise models and comparison to lysimeter data"

_Hydrology and Earth System Sciences, 2020_

## Referee Comment (RC1) · Rafael Schäffer (Referee) · 12 Oct 2020

1) General comment

Scientific significance

The topic of this manuscript is fitting well to the scope of HESS. The title is chosen appropriately (consider to substitute "Estimating" by "Estimation of", please). My only, but massive, point of criticism of this manuscript concerns the scientific significance. Neither the abstract (cf. my comment on line 1-3) nor the two final sections (cf. my comment on line 469ff) give attention to the scientific significance in a proper way.

[Figure]

What is the general value of this research? What is the real-world problem? What is interesting to people inside and outside this field? What about it is new? What is the specific problem, what do the authors achieve? After reading the whole manuscript this remains largely unclear to me. These questions should be addressed within the abstract, conclusion and maybe also partially within the introduction.

Scientific quality

The scientific methods and assumptions appear valid and clearly outlined to me. The results are convincing and support the interpretation. The description of the chosen approach is sufficiently complete, documented by several equations. As the authors provide the model code and software, fellow scientists are not only able to reproduce the results, but can do further developments as well. The authors give proper credit to related or previous works. The number and quality of references appears appropriate to me.

Presentation quality

Besides section 5.4, the overall presentation is well structured and clear. The English is easy to read and to understand. The manuscript is nearly free from technical mistakes. I have only two major remarks concerning abbreviations and units (see comment on line 80 and figure 7 in technical corrections). Some minor suggestions are given to improve the readability of some figures.

2) Specific comments

Lines 1-3: The abstract starts with a description of the specific topic already. Begin the abstract with two, three sentences tackling the large general topic of wide interest and the importance ("real world problem") of this study like you did in lines 16/17. Interest readers which are not that familiar with the topic might need this short introduction.

Lines73-74: Where is the research station Wagna located exactly? Refer to a town, city or valley, maybe draw a small map. Please provide some more geographic and climatic

information like altitude, slope (if applicable), annual average temperature, sunshine hours etc. As you state in lines 64-66, your idea is to provide a proof-of-concept with a single location. A good description is needed to evaluate how representative this location is and if this location is comparable to other sites.

Figure 1: Add the used symbols to the caption (e.g. precipitation p...) or describe the subfigures a, b, c and d separately.

Lines 83-89: This paragraph should be placed earlier in this chapter.

Figure 3: Also include $E_{t,s}/E_{max}$ and R in the caption.

Lines 267-268 and 279-284: The linear and non-linear models have eight, the lysimeter model five parameters. Did you check the influence of parameter variations? I could not get/understand, if you performed a sensitivity analysis (Figure A3?). Some parameters might be more important than others. It would be interesting to know if the variation of some parameters can be neglected, because they have a rather low influence on the model results.

Figure 6: What about using in subfigures b, c and the same colours for linear, non-linear and lysimeter as in subfigure a? I would prefer to use the same scale for the y-axis, even if this makes the figure a little larger.

Lines 390-391: Does it make sense to introduce a crop depended correction factor to receive better fits? Could the specific plant behaviour be handled similar to a hydrological process (cf. lines 429-430)?

Lines 469ff: I would be happy to see the title of this section renamed. What about "Relevance of this study", "impact" or "outlook". The latter would make more sense if matched with the conclusion. You are giving short outlooks in lines 416-417, 442-446 and 468 as well as a final outlook in lines 505-509. Wouldn't it be better to condense these statements in one paragraph at the end? What about the over-all benefit of your research? I have not the impression that you are bringing your

points across very well. Especially in the light of the two sentences in line 470 ("only one of many similar alternatives") and 472/473 ("other comparable non-linear model setups... perform in a similar manner"): Explain, what is the relevance of your research? What is new/special/innovative? What problem is solved/improved? Sell you product, sell your research in a better way! Ok, there is one statement ("TFN model improved...significantly") in line 496-497, but why is this important?

Line 513: References to the Durbin-Watson test and the Ljung-Box test seems appropriate to me here.

3) Technical corrections

Line 80 and elsewhere: This is maybe more a matter of taste, I prefer to us "a" instead of "yr".

Lines 84/85: "±2.5 m" in one line

Figure 2: Wouldn't it be better to place figure 2 within section 3.3?

Figure 3: The font size is rather small.

Line 244: DW = 2

Figure 4: Enlarge the size of this figure. Caption: Durbin-Watson (DW) statistics...

Line 288: one day

Lines 304, 308, 309, caption table 1 and elsewhere: Sometimes you write "10" and sometimes "ten". I would appreciate if you could unify this.

Figure 6: The position of the box is a little bit unfavourable. Describe the box content shortly in the caption.

Figure 7: "a" or "yr" instead of "y"

Figure 8: Your colour code for the lysimeter was red-brown in the previous figures. I suggest that you maintain the same colours.

Line 400: one

Line 451: delete "of"

Lines 461-463: Rephrase one of the sentences to avoid three times "the" at the beginning.

Figure A1: Caption: add "(ACF)" after autocorrelation function.

---

## Author Comment (AC1) · 28 Oct 2020

Dear Rafael Schäffer,

We thank the Reviewer for his review and constructive comments on the manuscript. In this short response we outline how we plan to address the most important issues that were raised in the review. A full response to the Reviewer will be provided at a later stage, awaiting the review from a second Reviewer and the Editors' decision.

Scientific significance

The major concern from the Reviewer is the scientific significance of this work and we

regret that this is not clear in the current manuscript. To address this important issue, we plan to add more explicit statements about this throughout the manuscript, in particular, as also suggested by the Reviewer, the Abstract, Introduction, and Conclusions. The main scientific contributions of this study are:

1. This study shows that non-linear TFN models can be successfully used to obtain estimates of groundwater recharge at different time scales, in areas where limited data is available: only groundwater levels, precipitation and evaporation are used.

2. For the first time, internal fluxes (e.g., recharge, evaporation) computed with the TFN model were compared to high resolution lysimeter measurements.

3. The use of non-linear TFN models improves the simulation of groundwater levels under drought conditions. This model should be preferred over the linear model, which is probably still the most-used model in practice.

4. The use of an ARMA(1,1) model can improve the capability of the noise model to reduce the residuals to white noise, and eventually pass diagnostic checks that allow for uncertainty quantification.

We will also highlight the significance of quantifying recharge for real world problems such as the adequate management of groundwater resources. The overarching research question is whether recharge can be estimated accurately from measured time series of groundwater levels, precipitation and potential evaporation. In this paper, a new method is developed to demonstrate that this is possible. Performance of the new method is assessed through comparison of the estimated recharge with recharge measurements of a lysimeter.

Description of the study site

We will add a figure with the geographic location of the research site and add more detailed information about the local conditions.

Technical Corrections

We agree with the suggested technical corrections from the Reviewer and will update the manuscript accordingly. We will make sure the use of units and numbers is consistent.

Improve discussion and conclusions

We plan to restructure parts of the Discussion and Conclusions according to the suggestions of the Reviewer to clarify these sections. Where applicable, we will add explicit statements on why certain findings are important and what their impact is. In particular, we will reconsider paragraph 5.4. The purpose of this paragraph was to clarify that (similar) non-linear models should be preferred over the linear model if one wants to estimate groundwater recharge or predict groundwater levels under drought conditions, because the hydrological processes are better represented. However, this comparison to the linear model is not clear from the paragraph and will be changed accordingly. We will also consider using this paragraph for highlighting the other aspects of the study that are relevant for a broader readership.

---

## Referee Comment (RC2) · Rodrigo Manzione (Referee) · 4 Dec 2020

The manuscript entitled "Estimating groundwater recharge from groundwater levels using non-linear transfer function noise models and comparison to lysimeter data" presents an interesting contribution on time series modelling (TSM) for hydrological purposes. Time-series modeling is an elegant way to treat monitoring data without the complexity of physical mechanistic models, still underused by the hydrological community.

I think the study is relevant and should be interesting for HESS readers. I agree with the first reviewer, the article is well written, well structured, the English is easy to read

and understand and the figures have quality. I compliment the flowchart: I have asked the authors of the works which I am invited to review to include when it is missing. Dr. Schäffer's review made my job easier, as he has already pointed out most of the points that I also noted and I will not keep repeating his words, but rather praise his review. Dr. Schäffer pointed out that my opinion should be the point on which the authors should pay more attention, which is the valuation of the work. What is it for? What is the advantage? Should I dedicate myself digest Hipel and McLeod (1994) text book and jump into time series models or master the modflow manual? There are several advantages about time series modelling that can be highlighted, Dr. Bakker is experienced with the subject and a paragraph about it can be easily incorporated.

Specific comments: - Introduction: highlight the problem and the advantages of TSM. Just an example: Line 35: "In recent decades, the use of a specific type of TFN models using predefined response functions (von Asmuth et al., 2002) has gained popularity for the analysis of groundwater levels (Bakker and Schaars, 2019)". Bakker and Schaars (2019) mention it, but if you present more studies, worldwide, with references from Australia, Brazil, Europe (there is a lot of studies in international journals with those cases studies), the readers could be convinced easily that it is one of the paths to follow. I recommend do add more references. And paint the whole picture about it (at least the last 10 years). - Study site and field data: a map of the study is welcome. Lysimeters as well, unless they are commercial as sounds like. - Software: is that available at GitHub? Are you publishing the code? It would be great, consider it. Section 4: I did not like the small graph under the others at Figure 6, too polluted. Section 5: the text of the items are too small to be individual items, consider changing the numbers (5.1, 5.2. . . ) by bullets. - Conclusion: too long, still with references, still sound loke discussion to me. Be mo direct to the point, staying just with the finds of your study and move back to the previous item the remaining text. - Appendix: I don't think the whole appendix is needed. The formulas and the test is described in the literature, just plots and tables are fine.

I would like to thank you the opportunity to comments on this paper, it was my first public discussion review and I think it is a very exiting procedure. I hope I had contributed to improve the paper in a constructive way and wish good lucky to the authors, reviews and editors in their carriers.

Best wishes,

Rodrigo Lilla Manzione, PhD. UNESP/FCE - Head of Biosystems Engineering Department Associate Professor - Water resources and geoinformation

---

## Author Comment (AC2) · 5 Jan 2021

Dear Rodrigo Manzione,

We would like to thank the second Reviewer for his review of our manuscript and the constructive comments to further improve our work. In this short response we outline how we wish to address the Reviewers' concerns in an updated version of this manuscript. With regard to the comments about the value of this work we acknowledge the importance of these comments here, and kindly refer to our response to the first Reviewer for how we plan to address this issue.

**Specific comments:**

The Reviewers' comments are in bold and our response in normal font.

**Introduction: highlight the problem and the advantages of TSM. Just an example: Line 35: "In recent decades, the use of a specific type of TFN models using predefined response functions (von Asmuth et al., 2002) has gained popularity for the analysis of groundwater levels (Bakker and Schaars, 2019)". Bakker and Schaars (2019) mention it, but if you present more studies, worldwide, with references from Australia, Brazil, Europe (there is a lot of studies in international journals with those cases studies), the readers could be convinced easily that it is one of the paths to follow. I recommend do add more references. And paint the whole picture about it (at least the last 10 years).**

We focused our literature review on the translation of precipitation and evaporation into groundwater levels fluctuations for this type of TFN models, instead of providing a complete literature review on the impulse response method (e.g., von Asmuth et al., 2002). We think this is appropriate for the presented work, but we agree with the Reviewer that adding a few case studies from around the world may convince more readers of applicability of these methods. We therefore plan to add citations to several case studies in a revised version of this manuscript.

**Study site and field data: a map of the study is welcome. Lysimeters as well, unless they are commercial as sounds like.**

A map of the study area and its location in Austria will be added.

**Software: is that available at GitHub? Are you publishing the code? It would be great, consider it.**

The software for the time series modeling (Pastas) is publicly available on GitHub (https://github.com/pastas/pastas), this includes all models and tests used in this study.

The scripts to analyze the data are available upon request, but since these require the non-publicly available time series data we did not make them public. The scripts are however available upon request from the first Author. We will add a statement in the "code and data availability" section about this.

**Section 4: I did not like the small graph under the others at Figure 6, too polluted.** We will consider adding a separate plot of the impulse and step responses for all three models in a revised version of this manuscript.

**Section 5: the text of the items are too small to be individual items, consider changing the numbers (5.1, 5.2. . .) by bullets.**

We plan to rewrite parts of Section 5 (we refer to our response to Reviewer 1) and will reconsider the subsection titles and numbering.

**Conclusion: too long, still with references, still sound loke discussion to me. Be mo direct to the point, staying just with the finds of your study and move back to the previous item the remaining text.**

We will consider moving parts of the Conclusions to the previous Section 5 and try to be more concise in the conclusions. We want to clarify that the references cited here are already cited earlier on in the manuscript and are only meant to show how the conclusions from this study are in line with findings in earlier studies.

**Appendix: I don't think the whole appendix is needed. The formulas and the test is described in the literature, just plots and tables are fine.**

We thank the Reviewer for this comment. We agree with the Reviewer that these tests are already well described in the literature and will remove this Appendix and add references to the original literature instead.

---

## Author Response (AR1)

Dear Editor, dear Reviewers,

We would like to thank the Reviewers Dr. Schäffer and Dr. Manzione for their reviews of our manuscript that allowed us to further improve the manuscript. We have revised the manuscript addressing all comments and suggestions made by the Reviewers. Most of the changes were made to address the comments from both Reviewers concerning the scientific significance of the presented work, most notably in the Abstract, and Sections one, five and six. Below we respond to each Reviewer individually, with the original comment in black and our response in green. All Changes that were made to the original manuscript are visible in the track changes document that is provided with the revision documents. Changes and additions are marked in blue and deletions are marked in red. All references to line numbers made in this response refer to the revised manuscript and not the track changes document. We have also made two minor changes to the manuscript (listed directly below) of two issues that we found during revisions.

Raoul Collenteur, on behalf of all Authors

Additional changes to the manuscript

**Drop the use of the R2 metrics and use RMSE instead**
To reduce the Python dependencies of the example scripts (which are now public) we dropped the use of the Hydrostats package for the computation of the goodness-of-fit metrics, and instead used the built-in metrics Pastas method. This change showed that the $R^2$ used for the previous version of the manuscript was different than normally defined, which should actually be similar to the Nash-Sutcliffe Efficiency (NSE). As the NSE and the $R^2$ were similar after this change we decided to remove the $R^2$ and use the RMSE instead to provide the reader with an additional goodness-of-fit metric.

**Allowing parameter Beta to be negative**
We made a minor change to the definition of the ARMA(1,1) noise model (Eq. (13)) after we received some comments on this (separate from this manuscript) on GitHub (https://github.com/pastas/pastas/issues/235). In the new definition the parameter Beta (line 268-270) is allowed to be negative as well, making the model more widely applicable. This change did not impact the estimated parameter values or any of the results presented in the manuscript.

**Response to Reviewer 1**

**1) General comment**

The topic of this manuscript is fitting well to the scope of HESS. The title is chosen appropriately (consider to substitute "Estimating" by "Estimation of", please). My only, but massive, point of criticism of this manuscript concerns the scientific significance. Neither the abstract (cf. my comment on line 1-3) nor the two final sections (cf. my comment on line 469ff) give attention to the scientific significance in a proper way. What is the general value of this research? What is the real-world problem? What is interesting to people inside and outside this field? What about it is new? What is the specific problem, what do the authors achieve? After reading the whole manuscript this remains largely unclear to me. These questions should be addressed within the abstract, conclusion and maybe also partially within the introduction.

We thank the Reviewer for his valuable comments and constructive feedback on our manuscript. We have adopted the suggested change in the title. To address the issue of scientific significance, we have added introductory sentences to the abstract (lines 1-2) to describe the real-world problem, and added a paragraph to the introduction section to introduce potential readers to the advantages of these methods (see also response to Reviewer 2).

The largest changes were made to Section 5 (now: Discussion) and 6 (Conclusions & Outlook), which were restructured and rewritten to address the comments from both reviewers. Section 5.1 now discusses the importance of our findings and non-linear recharge models in general. We have added statements throughout the manuscript to more clearly state the advantages of the method (e.g., lines 443-445: "*This may be particularly important when using this type of model to forecast groundwater recharge and levels under drought conditions.*"). We advocate for a more widespread use of non-linear TFN models, rather than a specific non-linear model, which is part of future investigations (stated now in lines 445-449). We now also discuss the challenges in the parameters estimation that arise when using non-linear models (lines 450-457).

We have also changed the Conclusions section to further highlight the novel aspects of this study (e.g., the use of lysimeter data, lines 508-510). The final paragraph of the conclusions now reiterates the advantages of the methods (lines 529-533) and recommends for more research into the suitability of different types of non-linear recharge models (lines 533-536).

**2) Specific comments**

Lines 1-3: The abstract starts with a description of the specific topic already. Begin the abstract with two, three sentences tackling the large general topic of wide interest and the importance ("real world problem") of this study like you did in lines 16/17. Interest readers which are not that familiar with the topic might need this short introduction.
Thanks for this suggestion, we added two introductory sentences to the abstract (lines 1-2).

Lines 73-74: Where is the research station Wagna located exactly? Refer to a town, city or valley, maybe draw a small map. Please provide some more geographic and climatic information like altitude, slope (if applicable), annual average temperature, sunshine hours etc. As you state in lines 64-66, your idea is to provide a proof-of-concept with a single location. A good description is needed to evaluate how representative this location is and if this location is comparable to other sites.
We agree with the Reviewer that a better description of the case study area was in place. We therefore added a Figure to the manuscript (Figure 1) which shows the location of the study area within Austria, and locations of the groundwater monitoring well, the lysimeters and the meteorological station used in this study. We also added a paragraph (lines 93-99) describing the local climate and, perhaps most importantly, why snow has not been considered in this study.

Figure 1: Add the used symbols to the caption (e.g. precipitation p…) or describe the subfigures a, b, c and d separately.

We added the symbols the caption of the Figure 2.

Lines 83-89: This paragraph should be placed earlier in this chapter.

We reordered the items in this chapter and placed the paragraph earlier in the text, see previous comments.

Figure 3: Also include Et,s/Emax and R in the caption.

We have added all the symbols used in the Figure 3 to the caption.

Lines 267-268 and 279-284: The linear and non-linear models have eight, the lysimeter model five parameters. Did you check the influence of parameter variations? I could not get/understand, if you performed a sensitivity analysis (Figure A3?). Some parameters might be more important than others. It would be interesting to know if the variation of some parameters can be neglected, because they have a rather low influence on the model results.

We did not perform a formal sensitivity analysis of the parameters for this study, (for example by fixing all parameters but one and looking at changes in the simulations) and only informally studied the parameter sensitivities using the standard errors and correlation between parameters (see for example, lines 232-233). Based on this analysis we found for example that it was possible to fix the parameter $S_{r,max}$ as it has a strong correlation with $k_s$. We agree with the Reviewer that a (formal) sensitivity analysis would be interesting, but we think this is outside of the scope of this manuscript. We have added a statement that no formal sensitivity analysis was conducted, but that the results from section 4.3 provide a good reason to do this in future research (lines 431-434).

Figure 6: What about using in subfigures b, c and the same colours for linear, nonlinear and lysimeter as in subfigure a? I would prefer to use the same scale for the y-axis, even if this makes the figure a little larger.

We tried changing the colours but found that it did not improve the Figure and made it more difficult to show the confidence interval. We therefore did not change the colours and only changed the scales for the y-axes (Figure 7 now).

Lines 390-391: Does it make sense to introduce a crop depended correction factor to receive better fits? Could the specific plant behaviour be handled similar to a hydrological process (cf. lines 429-430)?

In the case study area, the type of crop cultivated was changed every year (see also Fig. 9), and, as information on the type of crop is available, we think it would make sense to add a crop dependent factor to account for this change in crop type. In natural systems / different areas such information is probably not available. We therefore decided to apply the model using the simpler but more widely applicable approach of a constant factor. The lack of information about the vegetation in different areas would probably make it difficult to incorporate plant behavior as a hydrological process. We thus have not made any changes to the manuscript in response to this comment.

Lines 469ff: I would be happy to see the title of this section renamed. What about "Relevance of this study", "impact" or "outlook". The latter would make more sense if matched with the conclusion. You are giving short outlooks in lines 416-417, 442-446 and 468 as well as a final outlook in lines 505-509. Wouldn't it be better to condense these statements in one paragraph at the end? What about the overall benefit of your research? I have not the impression that you are bringing your points across very well. Especially in the light of the two sentences in line 470 ("only one of many similar alternatives") and 472/473 ("other comparable non-linear model setups… perform in a similar manner"): Explain, what is the relevance of your research? What is new/special/innovative? What problem is solved/improved? Sell you product, sell your research in a better way! Ok, there is one statement ("TFN model improved…significantly") in line 496-497, but why is this important?

We thank the Reviewer for this comment and agree that the previous version of the manuscript was not clear enough on these points. We have placed the text passage (lines 469ff in the previous

manuscript) within a new, larger section at the beginning of section 5, which discusses the relevance of non-linear models in a broader context. For our response to the other comments, we refer to our answer to the general comment above.

Line 513: References to the Durbin-Watson test and the Ljung-Box test seems appropriate to me here. We have removed this description from the Appendices (suggested by Reviewer 2 and we agree) and added references for both tests in the main text (lines 260-261).

3) Technical corrections

Line 80 and elsewhere: This is maybe more a matter of taste, I prefer to us "a" instead of "yr". We have updated the manuscript to use "a" everywhere.

Lines 84/85: "_2.5 m" in one line
This was an issue with Latex, change made.

Figure 2: Wouldn't it be better to place figure 2 within section 3.3?
Figure placement was chosen by Latex here, but it should indeed be in section 3.3.

Figure 3: The font size is rather small.
Font size was increased.

Line 244: DW = 2
Change made.

Figure 4: Enlarge the size of this figure. Caption: Durbin-Watson (DW) statistics…
Change made.

Line 288: one day
Change made.

Lines 304, 308, 309, caption table 1 and elsewhere: Sometimes you write "10" and sometimes "ten". I would appreciate if you could unify this.
All integers below 10 are now written as words, all number of 10 and more as integers.

Figure 6: The position of the box is a little bit unfavourable. Describe the box content shortly in the caption.
As we also received comments of this from the second Reviewer, we removed the box and added it as a separate Figure (Figure 8, described in lines 345-347).

Figure 7: "a" or "yr" instead of "y"
Change made.

Figure 8: Your colour code for the lysimeter was red-brown in the previous figures. I suggest that you maintain the same colours.
Good suggestion, this was indeed inconsistent. Change made.

Line 400: one
Change made.

Line 451: delete "of"
Change made.

Lines 461-463: Rephrase one of the sentences to avoid three times "the" at the beginning.
Change made.

Figure A1: Caption: add "(ACF)" after autocorrelation function.
Change made.

**Response to Reviewer 2**

As the concerns about the scientific significance of Reviewer 2 were shared with Reviewer 1, we refer to our response to the first Reviewer about how to manuscript was changed to address these issues. Below is our response to the specific comments from the second Reviewer.

Specific comments:

Introduction: highlight the problem and the advantages of TSM. Just an example: Line 35: "In recent decades, the use of a specific type of TFN models using predefined response functions (von Asmuth et al., 2002) has gained popularity for the analysis of groundwater levels (Bakker and Schaars, 2019)". Bakker and Schaars (2019) mention it, but if you present more studies, worldwide, with references from Australia, Brazil, Europe (there is a lot of studies in international journals with those cases studies), the readers could be convinced easily that it is one of the paths to follow. I recommend do add more references. And paint the whole picture about it (at least the last 10 years).

We thank the Reviewer for this comment and his suggestion to add more examples and references. We have added a paragraph to the Introduction with examples of case studies around the world (e.g., India, UK, Italy, Brazil, and later in the paragraph the Netherlands and Australia) and elaborated on the advantages of the impulse response method (lines 38-44). We think this provides future readers with a better understanding and appreciation of the methods presented this manuscript and why these should be considered to solve groundwater problems.

Study site and field data: a map of the study is welcome. Lysimeters as well, unless they are commercial as sounds like.

Good suggestion. We added a map of the case study area to the manuscript (Figure 1). Unfortunately, we are not allowed to publish most of the underlying data except the groundwater level time series (see also the following comment). The other data remains available for research purposes upon request from JR-AquaConsol.

Software: is that available at GitHub? Are you publishing the code? It would be great, consider it.

All methods and models were implemented in the Pastas Software (see also section 3.7) and are freely available on GitHub (https://github.com/pastas/pastas). We did not publish the original scripts to run the models and make the figures, as we are not able to publish the data necessary to run these scripts. We agree with the Reviewer that publishing the data and scripts would be great and have worked out a partial solution to publish at least some form of the scripts.

As an alternative, we published example scripts on how to use the presented methods with different but open data on Zenodo (https://doi.org/10.5281/zenodo.4548801) and Github. The groundwater levels are the same as used for this study and the precipitation and potential evaporation time series were obtained from the E-OBS database. We added a statement about this supplementary material in the 'Code and data availability' section after the conclusions.

Section 4: I did not like the small graph under the others at Figure 6, too polluted.

This comment was shared with the first Reviewer, and we decided to remove the graph from the original Figure and place it in a separate plot (Figure 8 in the revised manuscript).

Section 5: the text of the items are too small to be individual items, consider changing the numbers (5.1, 5.2…) by bullets.

We have restructured Sections 5 and 6 and changed some of the subheadings. We refer to our response to Reviewer 1 and the track changes document for all the changes that were made in this section.

Conclusion: too long, still with references, still sound loke discussion to me. Be mo direct to the point, staying just with the finds of your study and move back to the previous item the remaining text.

We have renamed this Section to 'Conclusions and Outlook' and made the two paragraphs with conclusions more concise. We have decided to keep to references in the conclusions, as we think it is important to make clear to the reader in the Conclusions that our findings confirm those from earlier studies, and why we advocate for a wider application of non-linear TFN models. We reiterate our statement given in the initial response that these references have been used earlier in the manuscript already and are not new.

Appendix: I don't think the whole appendix is needed. The formulas and the test is described in the literature, just plots and tables are fine.

We thank the Reviewer for this comment. We agree that this was redundant and have removed the appendix from the manuscript. Instead, we added references to these statistical tests to the main text body (lines 260-261).